# How Social Media Comments Inform the Promotion of Mask-Wearing and Other COVID-19 Prevention Strategies

**DOI:** 10.3390/ijerph18115624

**Published:** 2021-05-25

**Authors:** Sarah N. Keller, Joy C. Honea, Rachel Ollivant

**Affiliations:** Department of Communication, College of Liberal Arts & Social Sciences, Montana State University Billings, Billings, MT 59101, USA; jhonea@msubillings.edu (J.C.H.); rachel.ollivant@gmail.com (R.O.)

**Keywords:** pandemic, Facebook, mask-wearing, COVID-19, social media, vaccine hesitance, health promotion, health messaging, content analysis

## Abstract

Current COVID-19 messaging efforts by public health departments are primarily informational in nature and assume that audiences will make rational choices in compliance, contradicting extensive research indicating that individuals make lifestyle choices based on emotional, social, and impulsive factors. To complement the current model, audience barriers to prevention need to be better understood. A content analysis of news source comments in response to daily COVID-19 reports was conducted in Montana, one of the states expressing resistance to routine prevention efforts. A total of 615 Facebook comments drawn from Montana news sources were analyzed using the Persuasive Health Message Framework to identify perceived barriers and benefits of mask-wearing. A majority (63%) of comments expressed barriers, the most common of which were categorized as either misinformation about the virus or conspiracy theories. Benefits (46%) of mask-wearing were articulated as benefits to loved ones or people in one’s community or saving hospital space. This paper analyzes the implications of low perceived threat accompanied by low perceived efficacy of mask-wearing to make recommendations for future prevention efforts.

## 1. Introduction

As COVID-19, an infectious disease spread by the coronavirus, spread through rural America and new infection numbers rose to unprecedented peaks, many Mountain West communities remained non-compliant with the safety measures necessary to curb the infection tide, such as mask-wearing and social distancing. Unlike other parts of the country, the Mountain West region was rarely under statewide public health orders to wear masks in public, socially distance, close businesses and schools, and cancel events; however, even in states that did have a mask mandate, such as Montana, masks were not often worn in many conservative, rural counties [1].

While Americans largely complied with original COVID-19 prevention efforts in the spring of 2020, as the pandemic dragged on, COVID-19 fatigue set in and the lifting of safety orders across the nation over the summer provided hope that the worst was over. So, when cases began to rise in the fall, as predicted by experts, resistance to prevention efforts also grew [2]. This was especially the case in less populated rural states in the Midwest and the Rocky Mountain West, where case counts had remained comparatively low in the first wave of the pandemic and the safety precautions were interpreted as unnecessarily restrictive. The spring and summer of 2020 saw numerous public protests opposing safety measures, including anti-mask protests in Montana [1]. By October, cases in states in the upper Midwest and the Rocky Mountain West were spiking, and by November, nearly every state in the nation had skyrocketed into unprecedented levels of infection [2]. Political disputes were raging over whether the safety measures were effective or appropriate. In July 2020, Montana Governor Steve Bullock’s office had “received nearly 800 calls against the mask directive…more than double the number of comments in support” [1]. Yet federal health agencies, like the CDC, insisted that mask mandates and limits on public gatherings were necessary to stem the tide. This study sought to investigate the reasons for reluctance to comply with COVID-19 health and safety precautions in one state in the Mountain West region and to identify possible approaches to improve compliance. Using Witte’s Persuasive Health Message (PHM) model [3], we investigated the barriers to compliance with COVID-19 safety precautions and suggest messaging approaches that might affect health behavior and encourage people to adopt preventive measures.

The important question involves what people in this region (and elsewhere) say about the pandemic safety protocols and what reasons they cite for their resistance. Understanding the arguments voiced for resisting COVID-19 prevention in the United States is critical to both maintaining the current decline in cases and limiting flareups in the future. Barriers to COVID-19 prevention behaviors have been documented in the literature; they range from denial of perceived personal risk and skepticism about the severity of the health threat to doubts about the effectiveness of safety mandates and personal financial or logistical inability to comply [4]. By examining a systematic sample of comments on Facebook, a social networking site popular with Americans, we explored the rhetoric surrounding COVID-19 as expressed in Montana social media. Content analysis, a systematic study of communication artifacts, such as Facebook comments, was used to understand what people are saying about COVID-19 prevention. This research takes a qualitative approach to content analysis that carefully examines what these social media conversations look like in Montana and creates opportunities to discuss how this rhetoric may influence real-world behaviors regarding COVID-19 precautions. It also spurs discussion on how health behavior theories can be useful in countering resistance to public mandates. Individuals who do not perceive a health threat to be both severe and personally relevant are less likely to comply with recommended behavior changes to avert any health threat [3]. In the PHM framework, Witte explained how individuals must also believe that the recommended behavior change will, in fact, avert the threat, and that it is a behavior change they perceive as feasible [3]. The data about COVID-19 retrieved in this study will be examined in light of the PHM framework to identify possible strategies for public health messaging. This paper aims to shed light on the following research objectives: What do people in Montana say about COVID-19-preventive behaviors? What reasons do people in Montana voice for their resistance to COVID-19-preventive behaviors? Do Montana comments echo the debate about COVID-19 that occurred on a national scale?

This paper is broken up into the following sections: first, a background section or literature review is presented on the common resistance to COVID-19 prevention and the challenges associated with individual behavior change for public health. Next, a methodology section describes our research design and analysis. Results are presented following the methods. Our findings and their implications are described in our discussion and conclusion section at the end.

## 2. Background

People’s mixed feelings and beliefs make any public health policy based on individual choice complex. People often may know what they should do but choose to act differently based on convenience, comfort, or social pressures. Ambivalence (i.e., feelings of conflict) toward the recommended behavior plays a role. As Witte noted, people’s fear of the hassle or dislike of the connotations of an action may prompt them to reject messages promoting behavior change and react against the messenger [5]. For instance, in the United Kingdom, Maio et al. found a backlash reaction to antiracism messages when people were ambivalent toward ethnic minority groups [6]. Keller et al. found backlash reactions occurred among men when presented with domestic violence-prevention messages that stereotyped men as perpetrators [7]. Similar backlash reactions have occurred in response to anti-littering messages in the United States [8]. With the politically driven responses to COVID-19 precautions, exploring the feelings of conflict related to the sources of these messages may help us better understand the resulting behaviors. However, the literature does not advise health messengers on how to promote safety practices, such as COVID-prevention behaviors, when competing with political or psychological forces operating to resist the required changes in daily behaviors.

### 2.1. Politics and Prevention

People do not always react to health and safety threats in a rational manner [9]. “For some, the uncertainty and subsequent fear arising from [violent acts] creates a ‘sense of powerlessness,’ causing them to feel ‘they are at the mercy of dangerous forces beyond their control.’ This emotional reaction can instigate seemingly irrational behavior” [9]. In a climate of fear, political factions or belief groups can subvert popular fears to turn against logical safety measures, as with COVID-19 [4].

Political affiliation has been directly correlated with adherence to safety behaviors, with Republicans reporting a lower frequency of wearing face coverings than all other partisan identities [10]. The Montana Treasure State Pre-Election Poll showed a large partisan split on perceived health impacts of the coronavirus, with 76% of Republicans not at all or only a little worried, while a majority of Democrats (65%) were moderately or very worried. Similarly, Montana Republicans stood alone with a majority (51%) opposing statewide face coverings. By contrast, 95% of Democrats supported such orders, with 64% support among independents and 62% among those with other partisan identities. Nationally, partisanship is also a driver of compliance. In a study of cell phone mobility, Democratic counties were found to be more adherent to governors’ Twitter calls for social distancing compared to Republican counties—regardless of the political affiliation of the governor [11]. During the second half of 2020 (covering the time period this paper’s research was conducted), preliminary data suggests that the political affiliation of a state’s governor may impact COVID-19 cases and death rates, with Republican-led states harder hit than states led by Democrats [12].

Montana, with its rugged individualist ethos, its geographical and ideological divides, and its recent shift even further into the red state category, serves as an ideal case study of political divisiveness and its consequences for public health.

### 2.2. Misinformation

Mixed media messages and public statements about both the severity of the health risks of COVID-19 and the tactics for prevention have undermined beliefs in the reality of the threat and effectiveness of recommended safety behaviors, such as mask-wearing, social distancing, contact tracing, and quarantining [13]. In a study of 110 websites disseminating information about COVID-19, Cuan-Baltazar et al. found that only 1.8% (*n* = 2) met the Health on the Net Foundation Code of Conduct [14]. The Health on the Net Foundation Code of Conduct (HONcode) for medical and health Web sites addresses one of the Internet’s main healthcare issues: the reliability and credibility of information [15]. First established in 1996, the current version has remained unchanged since April 1997. This version is the result of a consensus amongst webmaster and key players. It has been translated and is in use in 35 different language versions. Further, more than 70% (*n* = 77) of health information websites had a low DISCERN health quality score, another validated rating tool that can be used by health professionals and the general public to assess the quality of health information contained on the Internet. Using 16 questions for assessing the reliability and quality of the consumer information, scored from one to five, DISCERN can be used to assign an index of the quality of information given by individual websites [16].

While exposure to online health information about COVID-19 from scientific sources, such as the World Health Organization and national health agencies, is correlated with increased preventive behaviors, the variability in quality-of-health information moderates this relationship [17]. Somma et al. found that beliefs in non-scientific information and conspiracy theories about COVID-19 were correlated with resistance to scientifically supported safety behaviors [18]. “Relying on social media and/or non-scientific websites as major sources of information on the COVID-19 is significantly and positively (albeit modestly) associated with the frequency of inadequate (i.e., non-scientifically supported) preventive behaviors, as well as with increased agreement with non-scientifically supported causal beliefs” [18]. The spread of misinformation has clearly promoted erroneous practices that have increased the spread of the virus [19]. These researchers called for renewed efforts to combat misinformation about the pandemic.

As a platform, Facebook is aware of the issues with health-related misinformation. While previous efforts by Facebook to control misinformation tended to stop short of deleting the content, as COVID-19 accelerated in America, the platform took aggressive measures to remove misinformation from the platform. In February 2021, Facebook announced an updated policy to remove any content promoting debunked information about COVID-19 [20].

These efforts focus on the original posts, but it is unclear how it will address misinformation posted in comments under credible sources. Currently, Facebook includes engagement as one of the standards for promoting comments; while they have other factors that attempt to screen low-quality comments, comments with the most reactions may be pushed to the top of a post [20]. This allows for commenters who are promoting misinformation to increase the visibility of the post based solely on the interactions with other users rather than the credibility of the information. A 30 November 2020 conversation between Facebook founder Mark Zuckerberg and infectious disease expert Dr. Antony Fauci showed a weakness in the comment-ranking system, as the attempt to give Dr. Fauci a platform to educate the public on COVID-19 was undermined by users actively countering Dr. Fauci and criticizing the information in the comment section. Because the dissenting comments spurred so much conversation and interaction, the threads were ranked as “Most Relevant” [21].

Considering the political divides related to COVID-19, we also see a stark divide in how Americans perceive a social media company’s efforts to control misinformation. A Pew Research study conducted in June 2020 discovered that “roughly three-quarters of U.S. adults say it is very (37%) or somewhat (36%) likely that social media sites intentionally censor political viewpoints that they find objectionable. Just 25% believe this is not likely the case” [22]. When adjusted for political views, 90% of Republicans believed that social media sites took political stances and “69% of Republicans and Republican leaners say major technology companies generally support the views of liberals over conservatives, compared with 25% of Democrats and Democratic leaners” [22]. More research is needed to identify effective strategies to combat misinformation about this pandemic and other topics.

### 2.3. COVID-19-Prevention Campaigns

Experts around the world have called for multi-lateral partnerships to reduce the spread of the virus. “The mass media, healthcare organization, community-based organizations, and other important stakeholders should build strategic partnerships and launch common platforms for disseminating authentic public health messages” [20]. Current COVID-19 messaging efforts by public health departments, community health services, and universities are primarily informational in nature and assume that audiences will make rational choices in compliance, contradicting extensive public health research indicating that individuals make lifestyle choices based on emotional, social, and impulsive factors [23,24,25,26].

Given the findings that resistance and misinformation about COVID-19 go hand in hand, public health and policy makers have advocated for technological changes to social media companies (e.g., Facebook, Twitter, Instagram, etc.) to launch data mining to detect and remove online content with no scientific basis from all social media platforms. “The only bastion of defense against rising public panic, financial market hysteria, and unintended misunderstandings of the science and epidemiology of severe acute respiratory syndrome coronavirus 2 (SARS-CoV-2) is agile, accurate, worldwide-available counter-information that takes the high moral ground and conveys a consistently science-driven narrative,” wrote Garrett near the start of the outbreak [13].

Another strategy is to research sources of resistance to social distancing and mask-wearing and to design campaigns accordingly. Theoretical strategies, such as the Health Belief Model (HBM), have been used to identify sources of resistance to recommended health behavior changes for a range of health threats [27]. Similar to the PHM framework, HBM suggests that individuals with high perceived barriers and low self-efficacy towards a recommended behavior change are less likely to make the change. In addition, individuals must perceive a health risk to be serious and personally relevant in order to adopt a change. The PHM brings several key concepts that are absent from HBM: (1) the concept of perceived response efficacy, i.e., the belief that a recommended behavior change will achieve its desired outcome; (2) the impact of culture, environment, and preferences on message response; and (3) the need to address audience demographics and psychographics in message design. These constructs have all been found to be relevant to COVID-19 behavior [28]. Therefore, the Persuasive Health Message Framework, which is based on HBM, was adopted as the guiding framework for this study.

Research that identifies potential predictors of engagement in preventive behaviors and testing will generate urgently needed insights into social responses to the pandemic and inform targeted interventions to promote preventive behaviors, testing, and vaccination compliance. Given the increased reliance on social media as a news source and the public concern over control over health messaging, understandings of how social media content shapes attitudes and behavioral intentions are crucial. This paper uses content analysis of Facebook comments in response to news stories about COVID-19 to shed light on the reasons people in Montana cite for their resistance to preventive behaviors:

RQ1: What reasons do people in Montana give for supporting or resisting COVID-19-preventive behaviors?

RQ2: Do Montana comments echo the debate about COVID-19 that occurred on a national scale, or are there unique sentiments expressed that reflect local demographics?

## 3. Methodology: Qualitative Content Analysis of Social Media Comments

Content analysis is a common method of inquiry in social science and mass communication research. Simply put, content analysis examines texts of various types to uncover themes related to the research questions. Traditional types of text included newspapers, magazines, television, and films. With the explosion of social media as a communication medium over the last decade, researchers have begun to mine comment boards, Twitter, and the most widely-used platform of all: Facebook. As of February 2021, Facebook had more users than any other platform, at 2.8 billion, and nearly 70 percent of American adults surveyed reported having a Facebook account [29].

There are myriad methods of content analysis. This study uses a qualitative approach, which involves the use of a category system to identify, classify, and analyze themes and patterns that emerge in the selected text [30]. Based on a list of pre-developed coding categories, we analyzed the content of 615 comments posted on the Facebook pages of statewide/regional/local news sites in response to daily COVID-19 updates during the fall 2020 surge of the pandemic, when public concern was rising about the spread of mis- and disinformation on social media.

### 3.1. Sampling Procedure

This study followed the guidelines developed by Franz et al. in their review of 23 studies that used Facebook for qualitative research. The researchers analyzed public user comments on Facebook in response to news outlets’ daily reporting of case counts in the state [31]. Though the daily case report was not the only pandemic-related news item posted each day, because every news organization in the state posted the report at about the same time each day, it allowed for consistent sampling. The sample included the public pages of three news outlets: NBC Montana [32,33,34,35], *The Great Falls Tribune* [36,37,38,39] and *The Billings Gazette* [40,41,42,43]. NBC Montana was selected because it is a statewide agency and has the largest Facebook following of any Montana news source. *The Billings Gazette* was selected because it is located in the largest city in the state and has the widest readership of any Montana newspaper. *The Great Falls Tribune*, located in a smaller city in central Montana, was added to capture comments that might focus more on local issues and concerns in comparison to the more statewide readership of the other two news organizations. This sampling gives us an opportunity to take a careful look at the conversation Montanans are seeing on social media about COVID-19.

Because this study used a passive data collection approach and did not recruit participants, it was not possible to determine the characteristics of the individuals who participated in the discussions. As noted by Franz et al., some studies mine the profile information provided by Facebook users to estimate the demographic characteristics of commenters [31]. However, it is not possible to verify the accuracy of public profile information, and Facebook users are free to generate any user profile they wish. Therefore, we did not attempt to ascertain the identities of those who participated in the discussions nor evaluate the relationship between user profile data and the comments posted. Instead, the comments themselves rather than the individuals serve as the unit of analysis for this study.

The readership of these news sources was selected based on the breadth (the wide scope of audience includes residents statewide) and range of political spectrum (while NBC Montana is perceived as a more left-leaning news source, the *Great Falls Tribune*, which reaches 13 counties in the center of the state, is considered to be more right-wing in slant). The *Tribune*, for example, has a hard copy circulation of 29,000 daily newspapers and 67,948 weekly readers online; the newspaper estimates it is read by 7 of 10 adults residing in its 117,900-resident region [44].

Individuals posting to the news source Facebook pages were believed to be those representing the strongest opinions with regards to COVID-19 on both ends of the political spectrum. “Even though the balance of opinions is not representative of the overall population, eavesdropping on these conversations shows how people feel and how they express their positions” [45]. In a comparison of Facebook viewpoints about climate change with a national study on how Americans feel about climate change, Kirk found that the social media source amplified the proportion of people dismissing climate science. She also found that the middle-ground viewpoints (people who were uncertain or unengaged) were more absent [45].

While automated bots and trolls are also a factor in any online conversation, their impact is hard to assess [46]. Recent steps by Facebook and Twitter to limit misinformation are credited with reducing the role of bots on those platforms, according to *The Washington Post* [47]. However, a 2018 study compared Twitter comments related to the Zika virus to survey data and found a relationship between the themes in the comments and the results of the survey, suggesting that a community’s social media activity may accurately offer insights into real-world attitudes [48]. For these reasons, analyses of the content of social media comments are an increasingly important tool in gathering information about public attitudes and behavioral intentions.

Since it is a relatively new methodological approach, there is little consensus on how to sample social media posts, and a wide range of methods have been used. For example, Takahashi et al. collected tweets regarding Typhoon Haiyan between 8 November and 13 November 2013 by searching tweets at three different time points each day [49]. Then, they created a study sample by taking a simple random sample of 1000 tweets. Other researchers used constructed week sampling [50,51]. To analyze tweets posted by journalists, Artwick randomly selected each day of the week from all available days during a certain date range [50]. Still others use a sample of convenience. For example, Greer and Ferguson [52] examined how local television used Twitter for promotion and branding. They first searched local television stations that managed Twitter sites on certain dates and found a total of 488 stations. They selected only the first page of each television station account because of the high volume of postings on sites. And finally, others search for the use of particular hashtags. Saura et al. searched for tweets using the #BlackFriday hashtag in the days just before and just after the event [53]. Additionally, Reyes-Menendez et al. selected all tweets that used “WorldEnvironmentDay” on one specific day [54]. This study combines several of these sampling techniques.

A consecutive-day approach was used to draw the sample. The sample period was the four-day span between 19 October and 22 October 2020. This date range was selected because it corresponded with rapidly rising cases in the state (the daily number of cases had more than doubled in the two weeks prior to the sampling period) and because tighter restrictions in the state’s most populated county had been enacted the previous week. Further, since weekdays tended to have the highest volume of responses, the researchers selected Monday through Thursday of the target week. The researchers sorted the entire population of comments using the Most Relevant feature on Facebook posts, which places the most active comments first, as determined by the number of reactions (clicks on the emojis expressing *like*, *care*, *angry*, etc.) and replies to each original comment. As such, those Most Relevant comments represent those that are seen by the most users. We then used the online software program ExportComments.com to capture and download the first 100 comments, including the nested sub-threads, in response to the daily case-count report for each of the three selected outlets on each of the four days included in the sample. The total number of comments varied significantly based on the number of individuals who commented and the number of replies to each initial comment, as nested replies were included in the sample.

### 3.2. Data Analysis Procedure

The comments were manually coded by two of the authors independently of each other, and inter-coder reliability was checked at the halfway point of analysis. Manually coding the comment using Excel spreadsheets populated with predeveloped categories allowed the researchers to look for nuance and subthemes within the comments. New categories were added as themes emerged. Comments were excluded if they were unrelated to the research questions. Comments in which the relevance was unclear were coded as such and reviewed by the research team for possible inclusion. Inclusion criteria were based on the research questions. At the broadest level, included comments were coded as related to either barriers to safety precaution compliance (specifically, mask wearing) and/or perceived benefits of compliance.

Using a priori coding categories based on the PHM framework while allowing other categories to emerge, one senior investigator and one graduate assistant read and coded the comments and made notes regarding their interpretations [3,15,55]. The entire team then compared and corroborated their coding and memos, returning when necessary to the original data, looking for themes and their interrelationships [55,56]. Discrepant interpretations led to in-depth discussions among research team members and ultimately to a richer understanding of the data. Through this iterative process, the team reached agreement on the themes pertaining to conceptual constructs found within the PHM: threat components (perceived threat of COVID); perceived benefits and barriers of preventive practices; perceived efficacy for prevention; and cues to action (including sources of information about COVID). Salient sub-themes within these components emerged in the analysis and are discussed in detail below.

## 4. Results

In sum, a total of 1108 comments were initially sampled (135 from the *Great Falls Tribune*, 603 from NBC Montana, and 270 from the *Billings Gazette*). Of these, 493 comments were excluded as irrelevant (e.g., memes, images, tags, links without context, etc.). Figure 1 shows the total number of included comments across the three sources was 615 (207 from the *Billings Gazette*, 77 from the *Great Falls Tribune*, and 331 from NBC Montana).

The following tables break down each category of coded comments on the COVID-19 updates. Comments may have been counted in more than one category if multiple themes were mentioned, putting the final percentages over 100%. As represented in Table 1, in response to RQ 1, 46% of the themes mentioned were supportive or in favor of taking the precautions to support community health, while 63% identified barriers to compliance, reflecting the divided opinions among Americans more generally and the pattern of resistance common among rural states in the Rocky Mountain West. We also coded for those who shared their personal experience with COVID as part of their argument for or against COVID precautions, but personal experience only made up 6.3% of the relevant themes, indicating that most commenters drew upon sources outside of their individual experience in formulating their views.

We then looked at the reasons cited for both compliance and non-compliance, to address RQ 1. While we had expected to see more comments related to cost concerns and efficacy as barriers given the abundance of national media coverage on the economic impact of shutdowns and the debates over the effectiveness of masks; these issues represented a minority of those expressing resistance to compliance. The confusion/misinformation (34.1%) and conspiracy (19.2%) categories represented more than half (53.3%) of the discussion among those identifying barriers. Because conspiracy and confusion/misinformation arguments tend to be closely connected, to distinguish between them, we only coded comments as conspiracy when they made a statement questioning the motivations behind the precautions. Those coded as confusion/misinformation only stated the incorrect information as motivation for their argument. The confusion comments also may have made a statement related to efficacy or cost concerns and, in those comments, the concern was driven by skepticism or confusion about the information that was encouraging the COVID-19 precautions. Table 2 lists the themes that were coded in the confusion/misinformation category, and Table 3 includes those coded as conspiracy categories.

One of the most concerning themes in the category of Confusion/Misinformation is that 11.5% (71) of the comments insisted that COVID-19 should be treated as just another cause of the death, and no special precautions ought to be taken. Further, those comments suggested that the focus on COVID deaths was leading to an unnecessary level of concern causing people to take unnecessary and unwarranted precautions. For example:


*I don’t doubt that COVID exists. I don’t doubt people are dying... BUT most “COVID deaths” that occur actually die from comorbidities!!! There are people dying from car accidents, shootings, and other reasons, also!! Where’s the breathless 24/7 Coverage of those?? The Lamestream media has y’all scared of your own shadows so the tyrranical [sic] govt can control you!!*
[42]

*I didn’t say they didn’t have it. I said they’re not sick. No symptoms. But their curiosity is making them get tested. So long as non-sick curious people keep getting tested, restrictions will not be lifted. This virus is going to do what viruses do: spread. We’ve already learned that mssks and lockdowns don’t work [sic]. Some of highest spread is coming from people’s private homes and gatherings, NOT in public places or work spaces*.[32]

These comments indicate a belief that the seriousness of the pandemic was being overblown both because of the media’s hyper-focus on reporting COVID-19 deaths and because increased testing meant more identified cases, even among those who did not feel sick.

This misunderstanding of case death rates and the seriousness of the virus was often accompanied by themes of fearmongering and people being sheep, which we categorized as conspiracy comments. For this reason, we coded the specific reasons commenters gave for those statements. Table 4 lists the specific claims cited when a comment was initially coded as conspiracy. Because some comments had more than one theme, the totals will go over 100%. Of the 113 total comments initially coded as conspiracy, nearly 60% (59.3%) identified generating fear in the public as a concern. Almost 32% specifically identified the media as the source and 7% specifically identified those following precautions as being sheep or adopting a herd mentality. Of particular concern were comments that referred to nefarious intentions on the part of government officials. Of the comments that were coded as conspiracy, over 60% mentioned political motivations, government dishonesty, or government overreach. This suggests that people’s beliefs that COVID-19 is not serious do not necessarily stem from simple misunderstanding or having inaccurate information, but are based on a general mistrust of elected officials and the media outlets that report COVID information.

For example, the following comment includes both an accusation of an agenda on the part of individuals reporting COVID case numbers and an unwillingness to believe the information one is given:

*I have empathy just no tolerance for liars [sic] who screw numbers to promote an agenda of control at all costs. If you can’t tell the truth about how someone died, how are we supposed to believe anything that you say is for the good of all. What I’m trying to say is believe half of what you see and none of what you hear. Best advice I ever got from my grandfather*.[42]

Due to the importance of masks in the COVID-19-prevention discussion, we also further analyzed all the mask-related comments to look for reasons for resistance. Table 5 summarizes the reasons commenters provided for their unwillingness to wear face coverings.

The data suggest that the commenters rejected wearing masks because they do not believe masks are effective in slowing the spread of COVID-19. Over half these comments (55.8%) question the legitimacy of the arguments for masks based on specific counter-arguments; 21.3% of these comments claim that areas with mask mandates still have serious outbreaks, 19.7% claim that they have seen science that says masks are not effective, and 14.8% said they know someone who wore a mask and still contracted COVID-19. For example, one individual who contracted COVID-19 even while wearing a mask in public concluded from her/his personal experience that masks are ineffective:

*This is my take on it. I social distanced only going to stores when needed [sic]. Masked up when I went out. Followed everything it says. Guess what I got on Saturday, I got the call cause I had went and tested due to symptoms, I tested positive for covid [sic]. No matter what if your going to get it, your going to get it [sic]. You can mask up do whatever you still will [sic]. Masks don’t help at all*.[42]

Another person made a claim about the lack of effectiveness of masks in preventing the inhalation of virus particles and cites their personal doctor as a supporting authority:


*That’s a barrier for large particles. But it will not stop microscopic particles of flu or common cold germs mister. Get with it here! Don’t be a fool! My doctor has been in practice for 35 years and is stronger in fighting this stupidity of masking then [sic] I am because she knows from years of training and practicing medicine that the mask thing is a freaking joke!*
[42]

Overall, the data indicate that there is significant resistance to compliance with COVID safety measures in Montana and that the reasons cited reflect the national debate, addressing RQ2.

## 5. Discussion

In response to RQ1, *what reasons people in Montana give for being for or resisting COVID-19-preventive behaviors*, this content analysis of discussion boards on news reports of case counts on Facebook found significant resistance to compliance with a statewide mask mandate (63 percent of relevant comments). Overall, the data indicate that social media comments express significant resistance to compliance with COVID safety measures in Montana. The most common reasons given for resisting safety measures were the beliefs that masks are not effective and suspicion that the severity of the pandemic was overblown by the media and/or government officials. Of particular concern was the prevalence of comments suggesting a conspiracy to mislead the public and enact government control over the citizenry. Individuals opposed to safety orders regularly rejected information about COVID-19 case numbers, suggesting that the data were either miscounted or deliberately falsified. Further, opponents to these measures routinely minimized the seriousness of the virus and suggested that “it’s like the flu” or that “almost everyone survives”.

As mentioned, those resistant to safety measures such as the use of face coverings frequently cited a belief that the precautions do not work. In addition to conspiracy theories about the government trying to control the public and criticisms of those complying (i.e., being sheep), opponents regularly argued that the fact that case counts were rising despite safety orders already in place was evidence that they simply do not work. Comments that indicated a fatalist approach were relatively common, with individuals concluding that there is no way to slow or stop the spread of the virus and thus there was no point in trying.

Results suggest that barriers to compliance with COVID safety precautions are related to both low perceived risk of COVID and low perceived efficacy of the prevention measures, two constructs that both HBM and PHM identify as necessary for behavior change. While public health campaigns could address those by highlighting the seriousness of the virus and providing information on the effectiveness of safety precautions, these findings raise questions about the fruitfulness of such efforts. Since a sizeable portion of the resistance to compliance seems to arise not from simple confusion or misunderstanding but from suspicion about conspiracies to mislead the public, health promotion campaigns must first confront the sources of these beliefs.

In response to RQ 2, *do Montana comments echo the debate about COVID-19 that occurred on a national scale*, these comments reflected the national debates over COVID-19 and reflect a lack of perceived threat from COVID-19 nationally. However, our findings depart slightly from the national data; the Montana Facebook commenters were more negative about safety precautions. In comparison with a survey of 1041 U.S. adults sampled nationwide in October 2020, our results show greater resistance to mask-wearing and lower perceived threat of COVID-19. The survey by the Center for Health and Safety Culture showed that 68% of respondents said they were very concerned about COVID-19, while approximately 63% of our comments indicated they thought the risk of COVID-19 had been overblown [57]. The national sample indicated that 55% of respondents always wear masks and more than 77% intended to wear masks in a public setting in the next seven days. By contrast, our Montana commenters indicated that 63% of commenters were opposed to wearing masks.

However, the skepticism in the national sample closely matched our findings. Whereas 38% of national respondents said they thought COVID was no different from the flu, 34% of our commenters voiced misinformation for reasons not to wear a mask. While 43% of national respondents said they thought government overreach was present in the response to COVID-19, only 19% of our commenters suggested government overreach (loosely related to conspiracy theories).

Our research concludes that, because of the reluctance among many residents to accept data and scientific evidence of the risks of COVID-19 and the effectiveness of prevention measures, health communication messages that seek primarily to inform individuals are likely to be ineffective. What is necessary are persuasive messages that appeal to groups suspicious of experts and resistant to government authorities. As suggested by PHM, audiences may need to hear from others similar to themselves culturally, demographically, and in terms of preferences. This might involve engaging role models from the target audience who may share similar personality traits but were able to put aside their doubts or convictions in favor of safety behavior that they came to believe would benefit their own communities or tribes. Modeling such a shift in perceived response efficacy would also fulfill one of the message constants recommended by PHM. Such role models, who presumably shifted from denial of risk to acceptance, may also be effective at raising the perceived threat among individuals in denial.

While that is a challenging task, this study does provide some hints of strategies that might be effective. The 46 percent of comments that supported COVID-19 safety precautions, like mask-wearing and staying home, mentioned benefits to people’s own health, the health of loved ones, or the protection of the community. Theoretically, even if polarization thwarts efforts to help the whole community, appeals to actions that benefit the individual or one’s in-group could have the potential to reach those reluctant to do their part in prevention. Emphasizing beneficial outcomes for loved ones may raise the perceived response efficacy among resistant audiences. Similarly, while no single preventive behavior will stop the virus, individuals who have low perceived efficacy may be persuaded by messages that place a greater emphasis on the collective effect of an array of preventive behavior.

The development of specific messaging requires further research, but this study indicates that public health messaging must move away from the presentation of scientific evidence without considering preconceived biases in the audience. With this in mind, messages designed to appeal to the general public to listen to health officials may be ineffective if it is based on the assumption that public health officials are considered credible in the community

When confronted by enclaves of misinformation, scientific messages might be interpreted as additional evidence of the attempt of the government to limit people’s freedom and to provoke a popular disdain of the elites by the average American. However, understanding where the audience stands in their relationship with politics and government systems, such as public health, may offer insights into ways to circumvent biases and guide the audiences to trust more credible health-related information. Further, understanding the factors that expose individuals to the risk of embracing misinformation is also crucial.

Our findings indicate that, when confronting a national threat that requires lifestyle behavior change on a societal level, the government may have a role in shaping norms about privacy and regulating the market of information. In the case of national emergencies of terrorism, pandemics, or disasters, societies will need to determine when sacrifices of personal freedom (or privacy) will be warranted for the end goals of safety and peace. To parlay this conversation, messages of urgency need to strike a balanced, unbiased tone—a tone that simultaneously communicates caution and invokes the values of independence and freedom. This approach holds the potential to reduce suspicions surrounding government control and overreach.

### 5.1. Limitations and Assumptions

By nature, social media offers limited information about who is behind the posts, and we have no demographic or background data on any of the users who contributed to this sample. When studying social media at a local level, it is necessary to consider what influence inauthentic profiles, such trolls and bots, may contribute to the conversation [46]. For this study, we were unable to determine the authenticity of the profiles that created the evaluated comments. While the study still reflects how COVID-19 attitudes are portrayed on social media, we are unable to determine if there may be any other influences in the conversations.

In addition, social media is only one method of seeking news and information and for discussing social and political issues. Social media is a growing and increasingly influential player in the creation and dissemination of cultural information. Nearly 75 percent of American adults use at least one social media source and Facebook is the most frequently used. But it draws a particular subset of the population: younger people, those with reliable internet access (urban dwellers), and those with more education [58]. These findings do not reflect the other kinds of conversations that individuals are having surrounding COVID-19 safety measures in their homes, at church, and in their local communities. The findings of this study are limited to how messaging could be developed for use within social media and cannot be generalized to other avenues for public health messages. Also, while social media can give us a glimpse into popular attitudes, the platforms tend to exacerbate polarization and extreme attitudes. Moderate voices tend to be lost in these online arguments. Other research models, such as surveys or focus groups, would help create a more well-rounded picture of Montana’s attitudes about COVID-19.

### 5.2. Further Research

These data offer a snapshot of four days during the 2020 COVID-19 outbreak as cases were climbing in Montana. In the weeks following the data collection, case counts continued to increase and affected more Montanans on a personal level. Since the beginning of 2021, COVID-19-related discussions began to focus more on vaccination availability and arguments for and against getting vaccinated. The Facebook comment boards may have shifted topic, but their role in informing public knowledge is likely to continue and to grow, as is the public debate over their impact and whether they should be censored or monitored. This study demonstrated the polarization of beliefs surrounding COVID-19 safety precautions and the specific reasons cited by Facebook users in one western state in shaping their decisions about whether or not to comply with local mandates and recommendations. This type of careful, qualitative analysis can be used by public health organizations to develop messaging that avoids exacerbating existing polarized views and is tailored to particular subsets within local or regional communities. Understandings of in-group and out-group values of are crucial importance to generate health behavior messages that do not alienate the individuals whose behavior one seeks to change nor encourage them to engage in behaviors that would ostracize them from their communities.

Further research on social media comment boards would also be useful in understanding how they shape actual decision-making and behaviors. Since this study did not survey individuals, we cannot draw conclusions about whether the individuals posting comments or reading along were affected by the discussions. We cannot determine whether their attitudes may have been reinforced or changed as a result of the debates or whether they engaged in any behavior modification. Later studies might evaluate individuals’ responses to social media comment boards and assess any attitudinal or behavioral changes because of consuming those messages. This would allow researchers to test whether carefully crafted messages have the intended effects.

Finally, additional research is needed on how social media platforms generally and comment sections specifically contribute to the political and social polarization discussed in this paper, especially in light of recent policy changes in which social media platforms increasingly label or remove content deemed inaccurate. While those actions could limit the spread of misinformation, they might also increase suspicion among those already distrustful of media and the government. The high number of comments in this study that referenced conspiracies to misguide the public should warn us to exercise caution in our public health messaging in the future. Additionally, without addressing the root causes of the U.S. trend toward tribalism and evaluating how social media either challenge or exacerbate existing divisions, the nation cannot begin to move toward closing the partisan gap and healing the rifts revealed by the COVID-19 pandemic.

## 6. Conclusions

U.S. cultural attitudes about the COVID-19 pandemic are sharply divided. The stark divide in attitudes on the comment boards in this study reflects that national trend. Just under half of the comments (46 percent) touted the benefits of observing safety precautions, while the rest indicated resistance to adopting such measures. This echoes the larger polarization of American society and has implications for controlling the virus and reducing the rates of death and serious illness. The denial of the existence of the pandemic, let alone its seriousness, has significant consequences for communities and their health care centers.

In the face of such ardent denial, health communication experts are hard-pressed to develop messages that will increase perceived severity and efficacy. Although campaigns based on the Health Belief Model [27] have shown success using factual messages in health promotion, the current atmosphere of distrust and anti-scientific sentiment have undermined the ability of health educators to use informational messages to fight the COVID-19 pandemic. It is likely that this tendency of individuals to reject official or expert messaging in favor of members of their own social enclaves will continue as long as social media echo chambers thrive. How to provide effective calls to communal action—be it for environmental conservation, health care, or actions important to preserving peace—is a question for future research. More needs to be done to identify health messaging strategies in this era of near-complete relativism and a decline in authenticity.

## Figures and Tables

**Figure 1 ijerph-18-05624-f001:**
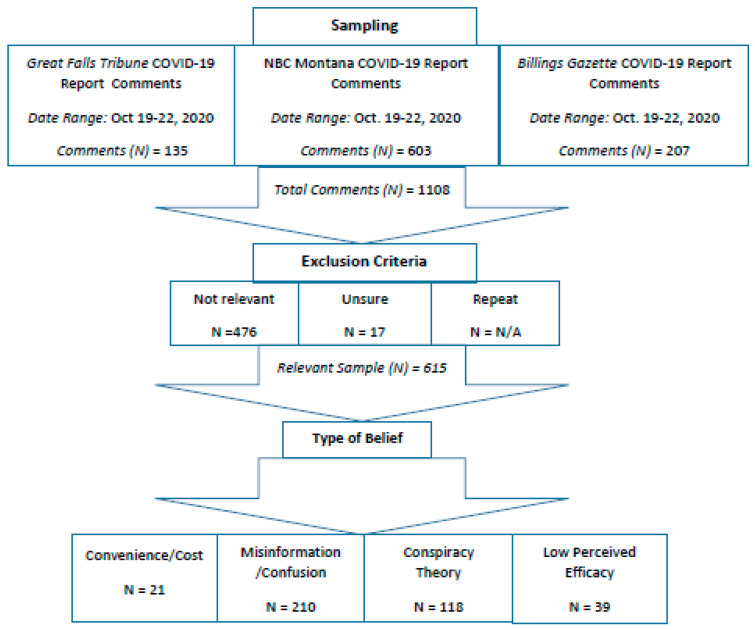
Sampling diagram of Facebook comments. Comments were sorted by the number of responses, from the most to least interactive comments. After sorting, the first 100 comments from each news source were taken each day of the week.

**Table 1 ijerph-18-05624-t001:** Totals of Categories.

Theme	Relevant Comments	
	*n*	%
Benefits (Supportive of precautions)	283	46
Personal Experience with COVID	39	6.3
Barriers (total)	388	63
Confusion/Misinformation	210	34.1
Conspiracy	118	19.2
Efficacy	39	6.3
Cost Concerns	21	3.41

Note. *n* = 615, including the total relevant comments in all three publications. Totals may exceed 100% as some comments were coded for more than one theme.

**Table 2 ijerph-18-05624-t002:** Themes related to confusion/misinformation.

Category	Relevant Comments	
	*n*	%
Skewed understanding of case rates	71	11.5
Belief that COVID-19 is low risk compared to other causes of death	59	9.6
Belief that masks are ineffective	57	9.3
Skewed understanding of death rates	17	2.8
Belief that masks do more harm than good	6	1.0

Note. *n* = 615, including the total relevant comments in all three publications.

**Table 3 ijerph-18-05624-t003:** Themes related to conspiracy.

Category	Relevant Comments	
	*n*	%
Taking COVID precautions is about fearmongering or being sheep	54	8.8
COVID-19 is over-hyped, too much media coverage	24	3.9
The Media is lying	12	2.0
Belief that cases and deaths are intentionally falsified	10	1.6
Government using COVID-19 to push agendas	7	1.1
Concern about government overreach/control	7	1.1
Hospitals and public health officials are lying to the public	4	0.7

Note. *n* = 615, including the total relevant comments in all three publications.

**Table 4 ijerph-18-05624-t004:** Specific claims found in the conspiracy comments.

Specific Claim	Total Relevant Comments	
	*n*	%
The media is creating fear around COVID-19	36	31.9
General statement about fear	31	27.43
Virus precautions are politically driven	26	23.01
Skewed death rates/overdiagnosis	23	20.4
Government fraud/dishonesty	23	20.4
Concern about government overreach/control	19	16.8
COVID-19 precautions are based on bad science	10	8.8
Mentions sheep or herd	8	7.1
States the importance of getting on with life	7	6.2

Note. *n* = 113, which is the total comments coded as Conspiracy.

**Table 5 ijerph-18-05624-t005:** Reasons for mask refusal.

Category	Relevant Comments	
	*n*	%
General “doesn’t work” statement	14	23.0
Masks do not slow the spread of COVID-19/areas with spikes also have mask mandates	13	21.3
Science does not back up mask wearing	12	19.7
Knows someone who wore a mask and got sick anyway	9	14.8
Waste of time/resources	5	8.2
Connecting masks to fear or being sheep	3	4.9
Other	3	4.9

Note. *n* = 61, which is the total comments mentioning masks.

## Data Availability

Not applicable.

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
