# Peer review of "How Social Media Comments Inform the Promotion of Mask-Wearing and Other COVID-19 Prevention Strategies"

_ijerph, 2021, doi:10.3390/ijerph18115624_

Round 1

Reviewer 1 Report

Thank you for giving me the possibility to review this article. I hope the authors find my comments productive and that they help them to improve their research work.

In this paper the authors analyse 615 comments from 3 Facebook profiles with the aim of analysing how Montanans perceive COVID-19 in relation to the use of mask-wearing and the threat of the virus as such.

Authors are encouraged to revise keywords and include mask-wearing and Facebook.

Authors are asked to revise the title of the article and adapt it to the topic under study: mask-wearing? The title should reflect what is to be found in the article.

In the introductory section, line 34 mentions "in that month", which month does it refer to? There are no references in the text above.

Authors are asked to expand on the information in the Introduction section, define the keywords when they are mentioned for the first time and put the object of study in context in a more extensive way.

Cite correctly line 55, 63, 98, 99, 104, 111, 207, 230, 347, 348, 376, 390, 393, 395, 558, 599, 604, 617.

The objective of the research should be clearly explained in the penultimate paragraph of the Introduction section and the last paragraph should explain the structure of the paper, listing the different sections.

Authors are asked to review when citing COVID and replace it with the correct term: COVID-19, and to unify when COVID-19 is mentioned with or without a hyphen, but always the same. It is also necessary to mention the author of the textual citation in lines 128 and 133 and the context of why this information is included.

The authors, in section 2. Background, should indicate the research gaps detected that have led them to carry out this research.

At the end of section 2, the research questions should be reflected and these should be clear and meaningful, so the authors are asked to remove the research questions from the methodology (section 3) and include them in section 2.

With respect to the literature review in section 2, a large part of it is devoted to focusing on COVID-19, the object of study of this paper, related to politics: political parties, political figures and personalities, which do not apply to the object of study to such an extent that such an extensive section should be devoted to them. Authors are asked to relate the theoretical framework to the subject matter of the paper.

Include HBM in brackets when first mentioning Health Belief Model as well as Persuasive Health Message.

One of the weaknesses of this research paper is the methodology. Unless the authors reinforce the method it cannot be considered for publication. The authors need to include a conceptual model asReyes-Menendez, A., Saura, J. R., & Alvarez-Alonso, C. (2018). Understanding# WorldEnvironmentDay user opinions in Twitter: A topic-based sentiment analysis approach. International journal of environmental research and public health, 15(11), 2537 or Saura, J. R., Reyes-Menendez, A., & Palos-Sanchez, P. (2019). Are black Friday deals worth it? Mining Twitter users’ sentiment and behavior response. Journal of Open Innovation: Technology, Market, and Complexity, 5(3), 58.

The presentation of the methodology is incoherent, it does not present a clear structure and, although the authors cite methodologies from other authors, it is not clear or explained on what basis they have decided to carry out the methodology of this paper. In this sense, the authors do not detail the software or system used to extract the data, Facebook comments in this case, and the coding of the comments is not justified since they cannot be coded independently as stated in line 416, nor can the analysis of the comments be carried out based on the criteria of the authors of this paper (as also mentioned).

Figure 1 is illegible. The authors cannot present a schematic with blue underlined words from the text editor they have used.

Authors must insert comments directly from Facebook with a brief comment at the beginning of the comments to put the reader in context. Full paragraphs of comments cannot be inserted as part of the results without an analysis of the comments or an explanation of why they are included (Line 486 and 491).

The authors make abusive use of inverted commas and citations.

There is no clear answer to the research questions posed through the results presented and after having analysed the data obtained. The results are not clear, nor are the conclusions of the study presented in a reasoned manner based on the results.

Furthermore, the authors do not compare their results with previous research results in order to rely on what other researchers have been able to prove before.

The authors are asked to create a separate Conclusions section and not to include them in the Discussion.

The references are current and correct although the authors are asked to improve them based on previous comments.

Author Response

May 7, 2021

Dear Reviewer 1,

The reviews on our Manuscript ID: ijerph-1187072 entitled “Motivating Montanans to Prevent COVID Now,” submitted to the International Journal of Environmental Research and Public Health, have been addressed below.  We feel the comments of the reviewer(s) greatly improved the paper. Please note, that due to extensive rewriting, the line numbers now only roughly correspond to the revised text. Therefore, we are submitting two versions – one with complete markup that corresponds to the line numbers in this letter, and a clean copy that includes more rewriting in response to the reviewer’ recommendations. Once our rewrite became too extensive to follow with markup, we switched to a new version. We hope the reviewers can follow our work, and find the new version acceptable for publication.

Reviewer 1:

  1. The first reviewer recommends the authors revise the keywords and include mask-wearing and Facebook. This is a good idea, and has been done.
  2. The authors are asked to revise the title of the article and adapt it to the topic under study: mask-wearing. This is also a good suggestion and has been done. The new title reads: How Social Media Comments Inform the Promotion of Mask-Wearing and Other COVID-19 Prevention Strategies.
  3. The month under study has been clarified in the introduction. The new lines 33-34 read: “The positivity rate in the “Mountain West” states, including Montana, remained relatively low over the summer and then spiked in September 2020, just when other states were experiencing a reprieve [1]. Total hospitalizations increased by one-third in that month for the region [2].”
  4. The authors are asked to expand on the information in the Introduction section, to define the keywords when they are mentioned for the first time and put the object of study in context in a more extensive way. This is an excellent suggestion. Some key words were defined in lines 26-19:

As COVID-19 – an infectious disease spread by the coronavirus - spread through rural America, and new infection numbers rose to unprecedented peaks, many Mountain West communities remained non-compliant with the safety measures necessary to curb the infection tide, such as mask-wearing and social distancing.

We have made the following changes to the Introduction, in lines 46-75:

The more important question, though, involves what people in this region (and elsewhere) say about the pandemic safety protocols and what reasons they cite for their resistance   Individuals are frequently required to make behavior changes for the collective good, be it for public health or national resilience and safety. Understanding the arguments voiced for resisting COVID-19 prevention and vaccines hesitancy in the United States is critical to both maintaining the current decline in cases and limiting flareups in the future. Barriers to COVID-19 prevention behaviors have been documented in the literature; they range from denial of perceived personal risk and skepticism about the severity of the health threat, to doubts about the effectiveness of safety mandates and personal financial or logistical inability to comply [5]. We hoped, that by examining a systematic sample of comments on Facebook, an American social networking site, we might begin to explore whether this rhetoric represents genuine concerns or if it merely echo national debates that have been voiced by political leaders and run amok on social media. Can regulating social media discourse limit popular buy-in to misinformation? Content analysis, a systematic study of communication artifacts, such as Facebook comments, was used to understand what people are saying about COVID-19 prevention, and what they say about where such arguments come from.

We then looked at whether health behavior theories can be useful in countering resistance to public mandates. Individuals who do not perceive a health threat to be both severe and personally relevant are….

  1. The authors are asked to correctly cite lines 55, 63, 98, 99, 104, 111, 207, 230, 347, 348, 376, 390, 393, 395, 558, 599, 604, 617. The citations have been corrected.
  2. The authors are asked to more clearly explain the research objectives in the penultimate paragraph of the Introduction section and to explain the structure of the paper in the last paragraph. Thank you for making this excellent suggestion. We have added the following text at the end of the Introduction section to clarify our research objectives and structure in lines 81-93:

This paper uses content analysis of Facebook comments in response to news stories about COVID-19 to shed light on the following research objectives: What do people in Montana say about COVID-19 preventive behaviors?: 1) What reasons do people in Montana voice for their resistance to COVID-19 preventive behaviors? Do Montana comments echo the debate about COVID-19 that occurred on a national scale?

This paper is broken up into the following sections. First, a background section or literature review is presented on the common resistance to COVID-19 prevention, and the challenges associated with individual behavior change for public health. Next, a methodology section describes our research design and analysis. Results are presented following the methods. Our findings and their implications are described in our discussion and conclusion section at the end.

  1. The authors are asked to review when citing COVID and replace it with the correct term: COVID-19, and to unify when COVID-19 is mentioned with or without a hyphen, but always the same. It is also necessary to mention the author of the textual citation in lines 128 and 133 and the context of why this information is included. Thank you for pointing out this flaw. All COVID-19 mentions have been updated for consistency. We have also added the following author attribution: “As Witte noted, people’s fear of the hassle or dislike of the connotations of an action may prompt them to reject messages promoting behavior change and react against the messenger [12]. Added “With the politically driven responses to COVID-19 precautions, exploring the feelings of conflict related to the sources of these messages may help us better understand the resulting behaviors” at the end of the paragraph.
  2. We are asked to indicate the research gaps detected in the Background that have led us to carry out this research. This has been done throughout the Background section: The following sentence was modified at line 103: However, the literature does not advise health messengers on how to promote safety practices, such as COVID prevention behaviors, when competing with political or psychological forces operating to resist the required changes in daily behaviors.

The following sentence was added to line 136: More research is needed to avert such backlash reactions, particularly when messaging to a resistant audience.

The following sentence was modified at line 150: Therefore, more research is needed to identify the root causes for resistance among specific subgroups is crucial for the development of effective message strategies targeted at that subgroup.

The following sentence was added to line 280-281:

More research is needed to identify effective strategies to combat misinformation about this pandemic and other topics.

  1. At the end of section 2, the research questions should be reflected and these should be clear and meaningful, so the authors are asked to remove the research questions from the methodology (section 3) and include them in section 2. This has been done in lines 379-383:

RQ1: What reasons do people in Montana give for supporting  or resisting  COVID-19 preventive behaviors?

RQ2: Do Montana comments echo the debate about COVID-19 that occurred on a national scale or are there unique sentiments expressed that reflect local demographics?

  1. With respect to the literature review in section 2, a large part of it is devoted to politics: political parties, political figures and personalities, which do not apply to the object of study to such an extent that such an extensive section should be devoted to them. The authors are asked to reduce the amount of literature devoted to politics and relate the theoretical framework more carefully to the subject matter of the paper. The discussion of political divisiveness was greatly pared down to focus only on its relevance to understanding the poliarized attitudes and behaviors around COVID-19.
  2. Include HBM in brackets when first mentioning Health Belief Model as well as Persuasive Health Message. This has been done.
  3. One of the weaknesses of this research paper is the methodology. Unless the authors explain the method it cannot be considered for publication. The authors need to include a conceptual model as Reyes-Menendez, A., Saura, J. R., & Alvarez-Alonso, C. (2018). Understanding# World Environment Day user opinions in Twitter: A topic-based sentiment analysis approach. International journal of environmental research and public health, 15(11), 2537 or Saura, J. R., Reyes-Menendez, A., & Palos-Sanchez, P. (2019). Are black Friday deals worth it? Mining Twitter users’ sentiment and behavior response. Journal of Open Innovation: Technology, Market, and Complexity, 5(3), 58. Thank you for the excellent suggestions. We have attempted to improve the explanation of the methodology and clarify our choice of qualitative content analysis as the data analysis approach. We have included examples of studies that used Facebook as the data source and explained differing approaches to data collection and analysis. We have included the recommended sources as evidence of varying ways to capture an appropriate sample using social media sources. We believe these suggestions have made the paper stronger.

  1. The presentation of the methodology is incoherent, it does not present a clear structure and, although the authors cite methodologies from other authors, it is not clear or explained on what basis they have decided to carry out the methodology of this paper. In this sense, the authors do not detail the software or system used to extract the data, Facebook comments in this case, and the coding of the comments is not justified since they cannot be coded independently as stated in line 416, nor can the analysis of the comments be carried out based on the criteria of the authors of this paper (as also mentioned). Thank you for pointing this out. We have changed the wording referencing our use of independent researchers to code the data, and added a line naming the softare used: “We then used an online program called ExportComments.com”

We also rewrote the methods section to make clear the approach used and the rationale for such an approach in lines 400-406. We included the name of the software program used and explained more clearly how the coding was conducted.

  1. Figure 1 is illegible. The authors cannot present a schematic with blue underlined words from the text editor they have used. We have reformatted the Figure for clarity.
  2. The authors must insert comments directly from Facebook with a brief comment at the beginning of the comments to put the reader in context. Full paragraphs of comments cannot be inserted as part of the results without an analysis of the comments or an explanation of why they are included (Line 486 and 491). We have cleaned up the results section to summarize significant findings, link the text with the data in the tables and provide context for the included quotes. We have added a summary at the end of the section:

“Overall, the data indicate that there is significant resistance to compliance with COVID-19 safety measures in Montana. They suggest that barriers to compliance with COVID-19 safety precautions are related to both low perceived risk of COVID-19 and low perceived efficacy of the prevention measures. While public health campaigns could address those by highlighting the seriousness of the virus and providing information on the effectiveness of safety precautions, these findings raise questions about the fruitfulness of such efforts. Since a sizeable portion of the resistance to compliance seems to arise not from simple confusion or misunderstanding, but from suspicion about conspiracies to mislead the public, health promotion campaigns must first confront the sources of these beliefs.”

  1. The authors make abusive use of inverted commas and citations. We apologize for this oversight and have removed all quotation marks that were not a direct quotes; not sure what to do about the citations
  2. There is no clear answer to the research questions posed through the results presented and after having analysed the data obtained. The results are not clear, nor are the conclusions of the study presented in a reasoned manner based on the results. The results have been clarified both in the Results section and the answers to the research questions have been added to the Discussion. In lines 640-643, we added:

“As represented in Table 1, in response to RQ 1, 46% of the themes mentioned were supportive or in favor of taking the precautions to support community health, while 63% identified barriers to compliance, reflecting the divided opinions among Americans more generally and the pattern of resistance common among rural states in the Rocky Mountain West.”

On lines 651-652, we added:

“We then looked at the reasons cited for both compliance and non-compliance, to address RQ 2.”

On lines 748-750, we added:

“Overall, the data indicate that there is significant resistance to compliance with COVID safety measures in Montana, and that the reasons cited reflect the national debate, addressing RQ 3.”

In the Discussion, in lines 873-885, we added:

“In response to RQ 1, this A content analysis of discussion boards on news reports of case counts on Facebook found significant resistance to compliance with a statewide mask mandate (63 percent of relevant comments). In response to RQ 2, the  with the most common reasons given for resisting safety measures were the being be-liefs that masks are not effective and suspicion that the severity of the pandemic was overblown by the media and/or government officials. Of particular concern was the prevalence of comments suggesting a conspiracy to mislead the public and enact government control over the citizenry. Individuals opposed to safety orders regularly re-jected information about COVID-19 case numbers, suggesting that the data were ei-ther miscounted or deliberately falsified. Further, opponents to these measures rou-tinely minimized the seriousness of the virus and suggested that “it’s like the flu” or that “almost everyone survives.” In response to RQ 3, These comments reflected the national debates over COVID-19 and reflect a lack of are evidence that there is a low level of perceived threat from COVID-19 nationally.”

  1. Furthermore, the authors do not compare their results with previous research results in order to rely on what other researchers have been able to prove before.

Please see our response to Reviewer 2’s no. 6 below.

  1. The authors are asked to create a separate Conclusions section and not to include them in the Discussion. This is an excellent suggestion, and, since the Discussion is so long, we have relabeled the Implications section as our Conclusion.

  1. The references are current and correct although the authors are asked to improve them based on previous comments. This has been done.

Reviewer 2 Report

Title: Motivating Montanans to Prevent COVID Now: How Social Media Comments Inform Discussions of Health Behavior

Manuscript Number: ijerph-1187072

International Journal of Environmental Research and Public Health (IJERPH)

The manuscript of “Motivating Montanans to Prevent COVID Now: How Social Media Comments Inform Discussions of Health Behavior” offers some insights for pandemic, health behavior, and social media. However, the research needs major revisions for publication in the IJERPH. Accordingly, I would like to provide the following suggestions:

  1. The title is not crisp and inviting. Authors need to state key novel insight in a single sentence if possible.
  2. Research issues, questions, and purposes are not fully developed. So, please build your research purpose and background based on prior studies.
  3. Regarding lines 25 and 28, the underlines are not necessary.
  4. In the line 348, “HPV” is first appeared in the text so authors should provide the full words.
  5. In the line 389, “anda” has a typo. Please revise it.
  6. The discussion section seems too brief so that it needs to be developed based on the findings of this research, along with comparing with the prior studies.
  7. The theoretical and practical implications are quite weak. Please improve academic and managerial contributions according to the findings of this study.
  8. With regard to Figure 1, it is not readable. Please make it clear.

Author Response

May 7, 2021

Dear Reviewer 2,

The reviews on our Manuscript ID: ijerph-1187072 entitled “Motivating Montanans to Prevent COVID Now,” submitted to the International Journal of Environmental Research and Public Health, have been addressed below.  We feel the comments of the reviewer(s) greatly improved the paper. Please note, that due to extensive rewriting, the line numbers now only roughly correspond to the revised text. Therefore, we are submitting two versions – one with complete markup that corresponds to the line numbers in this letter, and a clean copy that includes more rewriting in response to the reviewer’ recommendations. Once our rewrite became too extensive to follow with markup, we switched to a new version. We hope the reviewers can follow our work, and find the new version acceptable for publication.

Reviewer 2 provides the following suggestions:

  1. The title is not crisp and inviting. Authors need to state key novel insight in a single sentence if possible. The title has been greatly improved. Thank you for the suggestion.
  2. Research issues, questions, and purposes are not fully developed. So, please build your research purpose and background based on prior studies. See response to similar comments from Reviewer 1, above.
  3. Regarding lines 25 and 28, the underlines are not necessary. This was an error and the underlines have been removed.
  4. In the line 348, “HPV” is first appeared in the text so authors should provide the full words. Good catch. This has been done.
  5. In the line 389, “anda” has a typo. Please revise it. Thank you for noticing. This error has been corrected.
  6. The discussion section seems too brief so that it needs to be developed based on the findings of this research, along with comparing with the prior studies. Thank you for the great suggestion. The Discussion has been revised to compare our results with prior studies. For example, we added this section:

In response to RQ 3, these comments reflected the national debates over COVID-19 and reflect a lack of are evidence that there is a low level of perceived threat from COVID-19 nationally. However, our findings do depart slightly from the national data; the Facebook commenters were more negative about safety precautions. In comparison with a survey of 1,041 U.S. adults sampled nationwide in October 2020, our results show greater resistance to mask-wearing and lower per-ceived threat of COVID-19. The survey by the Center for Health and Safety Culture showed that 68% of respondents said they were very concerned about COVID-19, while approximately 63% of our comments indicated they thought the risk of COVID-19 had been overblown. The national sample indicated that 55% of respond-ents always wear masks and more than 77% intended to wear masks in a public set-ting in the next seven days. By contrast, our Montana commenters indicated that 63% of commenters were opposed to wearing masks. 

However, the skepticism in the national sample closely matched our findings. Whereas 38% of national respondents said they thought COVID was no different from the flu; 34% of our commenters voiced misinformation for reasons not to wear a mask. While 43% of national respondents said they though government overreach was present in the response to COVID-19, only 19% of our commenters suggested government overreach (loosely related to conspiracy theories).

To link the findings to national studies of misinformation, the following lines were added to the end of the Conclusion:

Scheufele and Krause (2018) found that individuals’ likelihood of being misinformed is is a function of a their ability and motivation to spot falsehoods, but also of group and social factors that increase the chances of citizens to be exposed to correct(ive) or corrosive information.

When confronted by enclaves of misinformation, scientific These messages are might be interpreted as additional evidence of the attempt of the government to limit people’s freedom and to provoke a popular disdain of the “elites” by the average American. However, understanding where the audience stands in their relationship with politics and government systems, such as public health, may offer in-sights into ways to circumvent biases and guide the audiences to trust more credible health related information. Further, understanding the factors which expose individuals to the risk of embracing misinformation is also crucial.

  1. The theoretical and practical implications are quite weak. Please improve academic and managerial contributions according to the findings of this study. Great point. The following paragraph in the Conclusion has been reworked to address the theoretical and practical implications:

In the face of such ardent denial, health communication experts are hard pressed to develop messages that will increase perceived severity and efficacy. Although campaigns based on the Health Belief Model [26] have shown success using factual messages in health promotion, the current atmosphere of distrust and anti-scientific sentiment have undermined the ability of health educators to use informational mes-sages to fight the COVID-19 pandemic. It is likely that this tendency of individuals to reject official or expert messaging in favor of members of their own social enclaves will continue as long as social media echo chambers thrive. How to provide effective calls to communal action – be it for environmental conservation, health care or actions important to preserving peace – is a question for future research. More needs to be done to identify health messaging strategies in this era of near complete relativism and a decline in authenticity.

To emphasize the practical implications, the following paragraph has also been added to the end of the Conclusion:

Our findings indicate that, when confronting a national threat that requires lifestyle behavior change on a societal level, the government may have a role in shaping norms about privacy and regulating the market of information. In the case of national emergencies of terrorism, pandemics or disasters, societies will need to determine when sacrifices of personal freedom (or privacy) will be warranted for the end goals of safety and peace. To parlay this conversation, messages of urgency need to strike a balanced, unbiased tone; a tone that simultaneously communicates caution and invokes the values of independence and freedom.

  1. With regard to Figure 1, it is not readable. Please make it clear. Figure one has been revised.

Round 2

Reviewer 1 Report

Dear authors, it is necessary to include the reviewer letter addressing the major concerns and summarizing the concerns with answers about the changes. 

Reviewer 2 Report

Title: How Social Media Comments Inform the Promotion of Mask-Wearing and Other COVID-19 Prevention Strategies

Manuscript Number: ijerph-1187072.R1

International Journal of Environmental Research and Public Health (IJERPH)

The revision of “How Social Media Comments Inform the Promotion of Mask-Wearing and Other COVID-19 Prevention Strategies” has been greatly improved. However, the authors need to revise several things for publication in the IJERPH. Accordingly, I would like to provide the following suggestions:

  1. Lines 353 & 356, the citations should be in numbers, but not in author names and year.
  2. Regarding the tables, please follow the format of the journal (refer to the template). The font seems too large and it should be 10 point. Please check the format and style of the journal and follow it.
  3. Regarding Figure 1, authors still need to improve more.
  4. Line 362, there is a typo (i.e., “et.”). Please check typos via the entire manuscript.
